# Hyperinsulinemic and Pro-Inflammatory Dietary Patterns and Metabolomic Profiles Are Associated with Increased Risk of Total and Site-Specific Cancers among Postmenopausal Women

**DOI:** 10.3390/cancers15061756

**Published:** 2023-03-14

**Authors:** Qi Jin, Ni Shi, Dong Hoon Lee, Kathryn M. Rexrode, JoAnn E. Manson, Raji Balasubramanian, Xuehong Zhang, Marian L. Neuhouser, Melissa Lopez-Pentecost, Cynthia A. Thomson, Suzanna M. Zick, Ashley S. Felix, Daniel G. Stover, Sagar D. Sardesai, Ashwini Esnakula, Xiaokui Mo, Steven K. Clinton, Fred K. Tabung

**Affiliations:** 1Interdisciplinary Ph.D. Program in Nutrition, The Ohio State University, Columbus, OH 43210, USA; 2Comprehensive Cancer Center, The Ohio State University, Columbus, OH 43210, USA; 3Department of Nutrition, Harvard T. H. Chan School of Public Health, Boston, MA 02115, USA; 4Department of Sport Industry Studies, Yonsei University, Seoul 03722, Republic of Korea; 5Department of Medicine, Brigham and Women’s Hospital and Harvard Medical School, Boston, MA 02115, USA; 6Department of Epidemiology, Harvard T. H. Chan School of Public Health, Boston, MA 02115, USA; 7Department of Biostatistics and Epidemiology, School of Public Health and Health Sciences, University of Massachusetts at Amherst, Amherst, MA 01003, USA; 8Division of Public Health Sciences, Fred Hutchinson Cancer Research Center, Seattle, WA 98109, USA; 9Sylvester Comprehensive Cancer Center, University of Miami Miller School of Medicine, Miami, FL 33136, USA; 10Mel & Enid Zuckerman College of Public Health, University of Arizona, Tucson, AZ 85721, USA; 11Department of Family Medicine, Michigan Medicine, University of Michigan, Ann Arbor, MI 48109, USA; 12Division of Epidemiology, College of Public Health, The Ohio State University, Columbus, OH 43210, USA; 13Division of Medical Oncology, Department of Internal Medicine, College of Medicine, The Ohio State University, Columbus, OH 43210, USA; 14Department of Pathology, College of Medicine, The Ohio State University, Columbus, OH 43210, USA; 15Department of Biomedical Informatics, College of Medicine, The Ohio State University, Columbus, OH 43210, USA

**Keywords:** dietary patterns, total cancer, insulinemia, inflammation, metabolomics

## Abstract

**Simple Summary:**

We investigated whether dietary patterns of insulinemia, inflammation and overall dietary quality are associated with the risk of total cancer, site-specific cancers, and pathological subtypes among postmenopausal women. We followed 112,468 women, 50–79 years of age, in the Women’s Health Initiative for a median of 17.8 years, documenting 18,768 incident invasive cancers. A higher overall dietary quality was associated with lower risk of total cancer and colorectal cancer. The potential of the dietary pattern to contribute to higher insulinemia and inflammation was associated with greater risk of total cancer, colorectal cancer and more strongly associated with risk of endometrial cancer and breast cancer (including triple negative breast cancer) than overall dietary quality. Additionally, a higher score of metabolites reflecting higher dietary quality was associated with lower lung cancer risk. Dietary patterns associated with cancer risk, therefore, warrant testing in clinical trials for cancer prevention among postmenopausal women.

**Abstract:**

We evaluated associations of the Empirical Dietary Index for Hyperinsulinemia (EDIH), Empirical Dietary Inflammatory Pattern (EDIP) and Healthy Eating Index (HEI2015) and their metabolomics profiles with the risk of total and site-specific cancers. We used baseline food frequency questionnaires to calculate dietary scores among 112,468 postmenopausal women in the Women’s Health Initiative. We used multivariable-adjusted Cox regression to estimate hazard ratios (HR) and 95% confidence intervals for cancer risk estimation. Metabolomic profile scores were derived using elastic-net regression with leave-one-out cross validation. In over 17.8 years, 18,768 incident invasive cancers were adjudicated. Higher EDIH and EDIP scores were associated with greater total cancer risk, and higher HEI-2015 with lower risk: HR_Q5vsQ1_(95% CI): EDIH, 1.10 (1.04–1.15); EDIP, 1.08 (1.02–1.15); HEI-2015, 0.93 (0.89–0.98). The multivariable-adjusted incidence rate difference(Q5_vs_Q1) for total cancer was: +52 (EDIH), +41 (EDIP) and −49 (HEI-2015) per 100,000 person years. All three indices were associated with colorectal cancer, and EDIH and EDIP with endometrial and breast cancer risk. EDIH was further associated with luminal-B, ER-negative and triple negative breast cancer subtypes. Dietary patterns contributing to hyperinsulinemia and inflammation were associated with greater cancer risk, and higher overall dietary quality, with lower risk. The findings warrant the testing of these dietary patterns in clinical trials for cancer prevention among postmenopausal women.

## 1. Introduction

Hyperinsulinemia and sustained inflammation are two proposed mechanisms driving cancer risk [1,2,3]. Dietary patterns that promote chronic hyperinsulinemia and chronic systemic inflammation may affect the risk of developing cancers and serve as modifiable risk factors for cancer prevention. We developed and validated two empirical hypothesis-oriented dietary indices: Empirical Dietary Index for Hyperinsulinemia (EDIH) and Empirical Dietary Inflammatory Pattern (EDIP), which predict the ability of the diet to contribute to insulin hypersecretion or chronic systemic inflammation, respectively. These dietary patterns are data-driven yet based on specific biological hypotheses relating diet with chronic disease [4,5].

EDIH and EDIP scores have shown stronger associations with cancer risk [6,7,8,9,10,11] than traditional dietary patterns in both men and women [12,13]. For example, dietary patterns including the Mediterranean diet and alternative healthy eating index, and other patterns, have not been consistently associated with cancer risk among women [14]. However, in the Nurses’ Health Study (NHS), higher EDIH was associated with a 22–47% higher risk of developing digestive system cancers [6,7]. Similarly, higher EDIP was associated with cancer risk among women in the NHS [8,15]. Due to advancing age and higher adiposity, postmenopausal women may represent a higher risk group for cancer related to these underlying mechanisms of malignant progression. However, outside of the NHS, these two dietary patterns have not been investigated in association with cancer risk among women.

Nutritional metabolomics may inform on more specific mechanistic pathways linking diet and cancer. Our metabolomics studies in the Women’s Health Initiative (WHI) suggested that patterns of cholesteryl esters(CEs), phospholipids, acylglycerols and acylcarnitines, may reflect the metabolic impact of insulinemic dietary patterns [16], while metabolites associated with coffee and lipid metabolism may reflect the metabolic potential of an inflammatory dietary pattern [17]. Among these metabolites, some have been evaluated for associations with risk of some cancers [18,19]. Although associations of EDIH and EDIP with cancer risk suggest that hyperinsulinemia and inflammation may broadly underlie these associations, the mechanistic pathways warrant investigation, and metabolomics profiles may provide a link to disease risk. The current study evaluated the etiologic role of EDIH and EDIP in relation to risk of total cancer, site-specific cancers and pathological subtypes, while comparing these associations with an established index of overall dietary quality—Healthy Eating Index-2015 (HEI-2015). We also characterized the plasma metabolomics profiles of each of the three dietary patterns and investigated their associations with cancer risk.

## 2. Methods

### 2.1. Study Population

The WHI enrolled 161,808 postmenopausal women aged 50–79 years between 1993 and 1998 in the United States. The study design has been described [20]. Briefly, the WHI study consisted of a three-component Clinical Trial (CT, n = 68,132) and Observational Study (OS, n = 93,676). The CT included a Dietary Modification trial (DM), two Hormone Therapy trials (HT), and a Calcium and Vitamin D trial (CaD). After the exclusions described in Appendix A, for each cancer site, we retained 112,468 women for the total cancer analyses. For the metabolomics aim, we used metabolomics data among 2306 participants from a matched case–control study in the WHI (BAA-24)—the Metabolomics of Coronary Heart Disease in the WHI [21]. The WHI protocol was approved by the institutional review boards at the Clinical Coordinating Center at the Fred Hutchinson Cancer Research Center (Seattle, WA, USA) and at each clinical center and all women signed informed consent. WHI is registered at clinicaltrials.gov as NCT00000611.

### 2.2. Dietary Assessment and Calculation of Dietary Indices

Dietary data from baseline—self-administered food frequency questionnaire (FFQ) representing intake in the preceding three months—were used to calculate the dietary indices [22]. The FFQ scanned data were processed with the University of Minnesota Nutrition Coordinating Center food and nutrient database (version 2005) to derive nutrient intakes [22,23]. The development and validation of the EDIH and EDIP scores have been described [4,5], and components of both scores in the WHI FFQ have been described as well [24]. The HEI-2015 measures adherence to the Dietary Guidelines for Americans (DGA) [25]. Appendix A shows the food group components of each index.

### 2.3. Ascertainment of Incident Cancer

Study outcomes included total cancer, invasive breast cancer (overall, ER+, ER-, PR+, PR-, HER2+, HER2-, ER- PR- HER2+, luminal A, luminal B, triple negative, invasive ductal carcinoma, invasive lobular carcinoma), colorectal cancer (colon, proximal colon, distal colon and rectum), non-cancerous intestinal polyps, endometrial cancer (overall, endometrioid, non-endometrioid), ovarian cancer (overall, serous, non-serous) and lung cancer (overall, small-cell, non-small cell). Primarily, CT and OS participants were contacted semi-annually and annually, respectively, to identify cancer diagnoses. Information on cancer incidence was initially verified by medical records and pathology reports and then underwent local and central adjudications by trained physicians [26]. Intestinal polyps were not adjudicated [26]. The definition of each cancer site is included in table footnotes and in Appendix A.

Time-to-cancer-development was defined as days from enrollment to the return of the follow-up questionnaire in which the event was reported. Participants were followed from enrollment to death, lost to follow-up or to the most recent follow-up (through 1 March 2019), whichever was first.

### 2.4. Metabolomics Profiling and Derivation of Metabolomics Profile Scores for the Dietary Patterns

The metabolomics profiling method used has previously been described [21]. Briefly, plasma metabolites were measured as peak areas using a targeted liquid chromatography tandem mass spectrometry (LC-MS/MS) metabolomics platform at the Broad Institute (Cambridge, MA, USA). The current study included a total of 509 named metabolites. Forty-five metabolites with >10% missing values were excluded. For 84 metabolites with <10% missing values, we imputed half the sample minimum value for the metabolite [16]. We transformed all metabolites using rank-based inverse normal transformation to achieve normal distribution of the metabolites [27]. We identified metabolomics profiles for adherence to each dietary pattern using elastic-net regression to regress each dietary index on the 464 metabolites, using a 7:3 training-to-testing dataset ratio and obtained the metabolomics signature using a leave-one-out cross-validation approach to avoid overfitting [28]. Metabolomics profile scores for each dietary pattern were derived from a weighted sum of metabolites selected via a series of elastic-net regressions and the weight for each metabolite was the regression coefficient of the selected model. Furthermore, we grouped metabolites into metabolomics class scores for each dietary index, using the pool of metabolites comprising each dietary index metabolomics profile score. Metabolomics class selection was determined non-empirically using information from the Human Metabolome Database (HMDB) to classify the metabolites into weighted metabolomics class scores.

### 2.5. Statistical Analysis

Each dietary index was adjusted for total energy intake using the residual method [29]. We used Cox proportional hazards regression to estimate hazard ratios (HR) and 95% confidence intervals (CI) of the relative associations of each dietary index and risk of developing total and site-specific cancers using the lowest dietary index quintiles as reference. Covariates included in the Cox models are listed in Appendix A. The proportional hazard assumption was assessed using Schoenfeld residual method and the time dependent covariate method. Because BMI [30] and type 2 diabetes [24,31] may strongly mediate the association of the dietary patterns and cancer risk, we additionally adjusted for these mediators in separate models.

In addition to the relative risk estimates, we calculated multivariable-adjusted absolute risk (incidence rate) of cancer in quintiles of each dietary index. Using the residual method [29], each dietary index was sequentially adjusted for each of the covariates included in the Cox models, then categorized into quintiles. Incidence rates per 100,000 person years were then calculated in dietary score quintile by dividing the number of cancer cases by the sum of the follow-up time within that dietary score quintile.

Metabolomics profile scores were categorized into tertiles because of the lower sample size and examined in relation to cancer risk (total cancer, colorectal, colon, breast, endometrial, lung cancers and intestinal polyps) using Cox regression and adjusting for the same covariates as in the diet analyses. Furthermore, we used Spearman correlation coefficients to assess correlations between metabolomics classes and dietary index food group components for each dietary pattern, adjusting for BMI, physical activity, and pack years of smoking.

Analyses were conducted using SAS version 9.4 (SAS Institute, Cary, NC, USA), and R Studio (2021.09.0) was used for data visualization. Two-sided *p* < 0.05 was considered statistically significant, and we further adjusted the nominal *p*-values for potential false discovery rate (FDR) using the Benjamini–Hochberg procedure.

## 3. Results

### 3.1. Participant Characteristics (Table 1)

Over a median of 17.8 years of follow-up, 18,768 incident invasive cancers were diagnosed. Participants who consumed the most hyperinsulinemic or most pro-inflammatory dietary pattern (EDIH/EDIP quintiles 5) or the lowest overall dietary quality per DGA (HEI-2015 quintile 1) were more likely to be Black or Hispanic or Latino, have a higher BMI and report lower physical activity and education levels.

**Table 1 cancers-15-01756-t001:** Distribution of participant characteristics in dietary index quintiles, using the total cancer analytic sample.

	Empirical Dietary Index for Hyperinsulinemic (EDIH) Score ^a,b^					Empirical Dietary Inflammatory Pattern (EDIP) Score ^a,b^					Health Eating Index 2015 (HEI-2015) ^a,b^				
Characteristic	Quintile 1	Quintile 2	Quintile 3	Quintile 4	Quintile 5	Quintile 1	Quintile 2	Quintile 3	Quintile 4	Quintile 5	Quintile 1	Quintile 2	Quintile 3	Quintile 4	Quintile 5
n	22,493	22,494	22,494	22,494	22,493	22,493	22,494	22,494	22,494	22,493	22,493	22,493	22,494	22,493	22,493
Race/ethnicity															
American Indian or Alaskan Native	0.3	0.3	0.5	0.5	0.5	0.4	0.3	0.4	0.5	0.6	0.6	0.4	0.4	0.3	0.3
Asian or Pacific Islander	2.2	2.8	3.2	3.1	2.7	1.3	1.9	2.3	3.5	5.1	2.7	3.1	2.9	2.9	2.5
Black	4.0	5.1	7.1	9.5	13	3.2	4.2	6.2	9.5	16	11	8.9	7.4	6.0	5.5
Hispanic/Latino	2.5	2.9	3.4	4.1	5.0	1.5	1.8	2.6	3.9	8.3	5.4	4.5	3.5	2.6	1.8
Other	1.4	1.5	1.3	1.4	1.5	1.2	1.2	1.4	1.6	1.8	1.6	1.5	1.4	1.3	1.3
White	89	87	84	81	77	92	91	87	81	68	78	81	84	87	88
Age, years	63 ± 7	64 ± 7	64 ± 7	63 ± 7	62 ± 7	63 ± 7	63 ± 7	64 ± 7	64 ± 7	62 ± 7	62 ± 7	63 ± 7	63 ± 7	64 ± 7	64 ± 7
BMI, kg/m^2^	26 ± 5	26 ± 5	27 ± 5	28 ± 6	30 ± 6	27 ± 5	27 ± 5	27 ± 5	28 ± 6	29 ± 6	29 ± 6	28 ± 6	27 ± 6	27 ± 5	26 ± 5
Under/Normal weight (15 ≤ BMI < 25)	49	44	39	34	24	44	42	39	36	29	29	33	37	42	49
Overweight (25 ≤ BMI < 30)	33	35	35	35	32	35	35	35	34	32	33	34	35	35	33
Obese (BMI ≥ 30)	18	21	25	31	43	21	23	26	29	39	38	32	27	23	18
Physical activity, MET-hours/week	17 ± 16	15± 14	13 ± 13	11 ± 12	9 ± 11	16 ± 15	14 ± 14	13 ± 13	12 ± 13	10 ± 12	8 ± 11	11 ± 12	13 ± 14	15 ± 14	17 ± 15
Pack years of smoking	11 ± 18	9 ± 17	9 ± 17	9 ± 18	11 ± 19	13 ± 20	10 ±18	9 ± 17	8 ± 17	8 ± 17	12 ± 21	10 ± 19	9 ± 18	9 ± 16	8 ±16
Current smoking	6	5	6	7	9	8	6	6	6	7	13	8	6	4	3
Aspirin/NSAIDs use	14	14	13	13	13	14	14	14	13	12	13	13	14	13	14
Statin use	2	2	2	2	2	2	2	2	2	2	2	2	2	3	2
Hypercholestrolemia	12	14	15	15	15	12	13	15	15	16	12	14	14	15	16
Educational level															
<high school	3	4	5	6	8	3	3	4	6	10	9	6	5	4	2
High school/GED	45	51	55	59	62	50	52	54	57	58	62	58	54	51	46
≥4 years of college	51	45	39	35	29	46	43	41	36	31	28	35	41	44	51
Total alcohol intake, alcohol servings/week ^c^	4.8 ± 7.5	2.4 ± 4.2	1.9 ± 3.7	1.6 ± 3.6	1.5 ± 3.6	5.3 ± 7.8	2.8 ± 4.4	1.9 ± 3.6	1.3 ± 3.0	0.8 ± 2.6	1.7 ± 4.0	2.2 ± 4.6	2.6 ± 5.0	2.8 ± 5.2	3.0 ± 5.5
Macronutrients, %kcal/d															
Carbohydrates	54 ± 10	54 ± 9	52 ± 8	49 ± 8	45 ± 9	50 ± 10	51 ± 9	51 ± 9	51 ± 9	50 ± 9	46 ± 9	48 ± 9	51 ± 9	53 ± 9	56 ± 8
Total fat	28 ± 8	29 ± 8	31 ± 8	34 ± 7	38 ± 7	30 ± 9	31 ± 8	32 ± 8	33 ± 8	34 ± 8	38 ± 7	35 ± 7	32 ± 7	29 ± 7	26 ± 7
Saturated fat	9 ± 3	10 ± 3	10 ± 3	11 ± 3	13 ± 3	10 ± 3	11 ± 3	10 ± 3	11 ± 3	11 ± 3	14 ± 3	12 ± 3	11 ± 3	9 ± 2	8 ± 2
Unsaturated fat	16 ± 5	17 ± 5	18 ± 5	20 ± 5	22 ± 5	18 ± 5	18 ± 5	19 ± 5	19 ± 5	20 ± 5	22 ± 5	20 ± 5	19 ± 5	17 ± 5	16 ± 5
Total protein	16 ± 3	17 ± 3	17 ± 3	17 ± 3	17 ± 4	17 ± 3	17 ± 3	17 ± 3	17 ± 3	17 ± 4	16 ± 3	17 ± 3	17 ± 3	17 ± 3	18 ± 3
Animal/plant protein ratio	2 ± 1	2 ± 1	2 ± 1	3 ± 1	3 ± 2	2 ± 1	2 ± 1	2 ± 1	3 ± 1	3 ± 1	3 ± 2	3 ± 1	3 ± 1	2 ± 1	2 ± 1
Micronutrients, per 1000 kcal															
Calcium, mg/d	577 ± 217	560 ± 211	528 ± 203	484 ± 184	410 ± 156	533 ± 205	531 ± 198	525 ± 201	511 ± 207	459 ± 201	419 ± 167	470 ± 183	508 ± 196	549 ± 204	614 ± 213
Potassium, mg/d	1860 ± 427	1816 ± 412	1728 ± 388	1599 ± 353	1384 ± 319	1882 ± 416	1781 ± 383	1709 ± 376	1616 ± 377	1397 ± 371	1296 ± 295	1523 ± 309	1688 ± 335	1848 ± 360	2030 ± 360
Vitamin D, mcg/d	3 ± 2	3 ± 2	3 ± 2	3 ± 2	2 ± 1	3 ± 2	3 ± 2	3 ± 2	3 ± 2	3 ± 2	2 ± 1	2 ± 1	3 ± 2	3 ± 2	3 ± 2
Magnesium, mg/d	180 ± 35	175 ± 35	167 ± 33	154 ± 30	134 ± 28	177 ± 34	170 ± 33	164 ± 34	158 ± 34	141 ± 35	126 ± 24	147 ± 25	162 ± 26	177 ± 28	197 ± 30
Iron, mg/d	8 ± 2	8 ± 3	8 ± 3	8 ± 2	7 ± 2	8 ± 2	8 ± 2	8 ± 3	8 ± 3	8 ± 3	7 ± 2	8 ± 2	8 ± 3	8 ± 3	9 ± 3
Folate, mcg/d	184 ± 64	183 ± 65	175 ± 64	161 ± 59	136 ± 50	184 ± 62	177 ± 61	171 ± 62	164 ± 63	142 ± 61	129 ± 52	152 ± 55	169 ± 59	185 ± 62	202 ± 62
Vitamin A, mcg RAE/d	483 ± 194	493 ± 199	486 ± 198	467 ± 197	423 ± 196	490 ± 205	485 ± 189	480 ± 188	470 ± 192	426 ± 210	385 ± 182	438 ± 188	469 ± 196	506 ± 192	553 ± 193
Vitamin C, mg/d	74 ± 40	76 ± 41	72 ± 39	63 ± 35	49 ± 28	70 ± 38	71 ± 38	70 ± 39	67 ± 38	54 ± 34	41 ± 26	57 ± 31	69 ± 36	79 ± 38	88 ± 38
Vitamin E, IU/d	6 ± 3	6 ± 4	6 ± 4	6 ± 3	5 ± 3	6 ± 3	6 ± 3	6 ± 4	6 ± 4	5 ± 3	5 ± 3	6 ± 3	6 ± 4	6 ± 4	7 ± 4

^a^ EDIP, EDIH and HEI2015 scores were adjusted for total energy intake using the residual method. Lower EDIP indicates anti-inflammatory diets, while higher EDIP scores indicate pro-inflammatory diets. Lower EDIH indicates anti-hyperinsulinemic diet while a higher score indicates pro-hyperinsulinemic diet. HEI-2015 measures adherence to the 2015–2020 Dietary Guidelines for Americans—higher HEI-2015 scores are indicative of greater adherence and higher dietary quality. ^b^ The values in the table are percentages for categorical variables and mean ± standard deviation for continuous variables. ^c^ Alcohol serving was the sum of beer (1 glass, 1 bottle or 1 can), wine (4 oz glass of red wine or white wine) and liquor (1 drink or 1 shot whiskey, gin, etc.).

### 3.2. Food and Nutrient Profiles of the Dietary Patterns (Appendix A)

There are 27 food groups comprising the EDIH and EDIP and nine are common to both indices, including red meat, processed meat, non-dark (non-fatty) fish, sugar-sweetened beverages (regular sodas), artificially sweetened beverages (diet sodas) and refined grains, contributing higher scores; leafy-green vegetables, wine and coffee contributed lower scores (Appendix A). Unique to EDIH are French fries, butter, margarine (high scores), whole fruit and whole dairy (low scores). The correlation between EDIH and EDIP was 0.63. In addition to foods, HEI has specific nutrients to reduce, such as saturated fats; therefore, the score only includes low/non-fat dairy foods. Additionally, HEI does not include foods without caloric value such as coffee or diet sodas. HEI-2015 had a correlation of −0.36 with EDIH and −0.26 with EDIP.

Participants consuming a more hyperinsulinemic dietary pattern consumed fewer calories from total carbohydrates and more calories from total fat, saturated fat, added sugar and animal protein. The macronutrient distribution (Appendix A) for EDIP was similar but with smaller contrasts between high and low EDIP compared to EDIH. Higher overall dietary quality based on higher HEI-2015 was characterized by higher intake of total carbohydrates and lower intake of total fat and saturated fat.

### 3.3. Metabolomics Profile Scores of the Dietary Patterns (Figure 1)

Of the 464 metabolites retained, elastic-net regression selected 93 for EDIH, 88 for EDIP and 67 for HEI-2015. Correlations among the metabolomics profile scores (m---) were mEDIH-mEDIP (0.73), mEDIH-mHEI-2015 (−0.61) and mEDIP-mHEI-2015 (−0.29). In addition, correlations between the metabolomics profile scores and their corresponding dietary scores were: mEDIH-EDIH (0.45), mEDIP-EDIP (0.33), and mHEI-2015-HEI-2015 (0.50). Higher EDIH was associated with higher amino acids and glycerophosphocholines (glyceroPC), and with lower mono/di-carboxylic acids (MCA, DCA), CEs, phosphosphingolipids (phosphoSL), alkaloids, purines, pyridines and pyrimidines (Figure 1). The metabolite classes were similarly associated with the EDIH food group components, e.g., amino acids correlated positively with red/processed meat and French fries, which contribute to higher EDIH, and inversely with fruit, leafy-greens, coffee and wine, which contribute to lower scores. EDIP had a similar pattern of correlations with the metabolomics classes. Alkaloids were strongly positively correlated with coffee, which contributes to lower scores in both EDIH and EDIP (Figure 1A,B). In contrast, alkaloids were not strongly correlated with HEI-2015, which does not include coffee. Higher HEI-2015 was correlated with higher DCA, indoles, benzoic acids, glyceroPE, glyceroPC, phosphoSL, purines, pyrimidines and pyridines and with lower carnitines, amino acids, TAGs and amines. However, saturated fat, a moderation component of HEI-2015 was also associated with higher carboxylic acids, glycerol-PE/PC/SL and lower carnitines and amino acids (Figure 1C).

**Figure 1 cancers-15-01756-f001:**
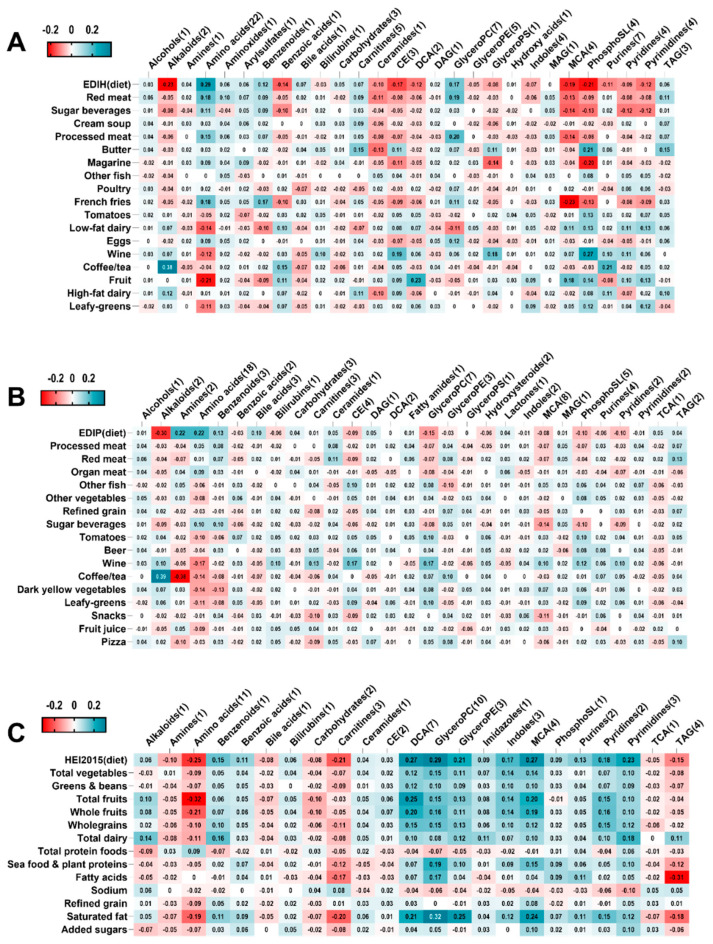
Correlations between dietary pattern food group components and metabolomics class scores. (**A**) Correlations between EDIH food group components and EDIH metabolomic class scores. (**B**), Correlations between EDIP food group components and EDIP metabolomic class scores. (**C**), Correlations between HEI-2015 food group components and HEI-2015 metabolomics class scores. Values are partial Spearman correlation coefficients adjusted for BMI, physical activity and pack years of smoking. EDIH, empirical dietary index for hyperinsulinemia; EDIP, empirical dietary inflammatory pattern; HEI-2015, healthy eating index-2015; TAG: Triradylcglycerols. MAG: Monoradylglycerols. DAG: Diradylglycerols. GlyceroPS: Glycerophosphoserines. GlyceroPE: Glycerophosphoethanolamines. GlyceroPC: Glycerophosphocholines. CE: Cholesterol esters. PhosphoSL: phosphosphingolipids. MCA: mono-carboxylic acids. DCA: di-carboxylic acids. TCA: Tricarboxylic acids.

### 3.4. EDIH and Cancer Risk (Table 2, Figure 2)

Women classified in the highest quintile of EDIH had greater risk of total cancer compared to those in the lowest quintile, with a multivariable-adjusted incidence rate difference of 52 per 100,000 person years and corresponding HR (95% CI) of 1.10 (1.04–1.15). Findings from the categorical analyses were aligned with EDIH modelled as a continuous variable (Table 2). A 1 sd increment in EDIH was associated with higher risk of colorectal cancer, colon cancer and proximal colon cancer but not distal colon or rectal cancer. EDIH was also strongly associated with intestinal polyps. Further, higher EDIH was associated with breast cancer and pathological subtypes including ER-negative, luminal B, invasive lobular carcinoma, and triple negative breast cancer. Higher EDIH was associated with a greater risk of endometrial cancer especially the endometrioid subtype, but not ovarian cancer or lung cancer (Figure 2). The EDIH metabolomics score was generally not significantly associated with cancer risk, but a 1 sd increment in the score was associated with elevated risk of endometrial cancer: HR, 3.58; 95% CI, 0.96–13.29 (Table 3).

**Table 2 cancers-15-01756-t002:** Multivariable-adjusted absolute and relative risk for the associations of dietary patterns and future development of total cancer, site-specific cancers and pathological subtypes ^a,b^.

Dietary Pattern	Cancer Risk Type	Quintile 1	Quintile 2	Quintile 3	Quintile 4	Quintile 5	Q5-Q1 (Absolute Risk Difference); ^c^ P-Trend. ^d^	FDR-Adjusted *p*-Value
**Total cancer (except non-melanoma skin cancer)**								
EDIH ^e^	Absolute risk/cases	969/3934	962/3861	993/3772	1015/3594	1021/3607	52	
EDIH	Relative risk	1 (reference)	1.04 (1.00, 1.09)	1.06 (1.01, 1.11)	1.04 (1.00, 1.10)	**1.10** **(1.04, 1.15)**	0.0008	**0.0122**
EDIP ^e^	Absolute risk/cases	979/4131	970/4027	992/3703	1000/3608	1020/3299	41	
EDIP	Relative risk	1 (reference)	1.06 (1.01, 1.11)	1.02 (0.97, 1.08)	1.07 (1.01, 1.13)	**1.08** **(1.02, 1.15)**	0.0163	**0.0808**
HEI-2015	Absolute risk/cases	1030/3697	1005/3740	967/3786	977/3750	981/3795	−49	
HEI-2015 ^e^	Relative risk	1 (reference)	0.98 (0.93, 1.02)	0.97 (0.92, 1.01)	0.94 (0.89, 0.98)	**0.93** **(0.89, 0.98)**	0.0008	**0.0122**
**Colorectal cancer**								
EDIH	Absolute risk/cases	80/3795	80/309	89/372	85/306	91/320	11	
EDIH	Relative risk	1 (reference)	1.06 (0.90, 1.25)	1.31 (1.11, 1.54)	1.08 (0.91, 1.28)	**1.19** **(1.00, 1.43)**	0.0658	0.2084
EDIP	Absolute risk/cases	84/315	82/333	82/338	86/319	91/303	7	
EDIP	Relative risk	1 (reference)	1.14 (0.96, 1.35)	1.19 (0.99, 1.43)	1.19 (0.98, 1.45)	**1.23** **(0.99, 1.52)**	0.0595	0.1966
HEI-2015	Absolute riskc/cases	96/342	86/337	86/316	76/297	82/316	−14	
HEI-2015	Relative risk	1 (reference)	0.96 (0.82, 1.12)	0.90 (0.77, 1.06)	0.83 (0.70, 0.97)	**0.86** **(0.72, 1.01)**	0.0158	**0.0808**
**Colon cancer**								
EDIH	Absolute risk/cases	67/250	66/264	74/312	72/254	76/266	9	
EDIH	Relative risk	1 (reference)	1.09 (0.91, 1.31)	1.33 (1.11, 1.60)	1.09 (0.90, 1.32)	**1.22** **(1.00, 1.48)**	0.0755	0.2295
EDIP	Absolute risk/cases	69/260	68/286	72/279	69/266	78/255	9	
EDIP	Relative risk	1 (reference)	1.18 (0.98, 1.43)	1.20 (0.98, 1.46)	1.22 (0.98, 1.51)	**1.28** **(1.02, 1.62)**	0.0483	0.1835
HEI-2015	Absolute risk/cases	80/285	71/284	73/264	64/249	67/264	−13	
HEI-2015	Relative risk	1 (reference)	0.97 (0.82, 1.14)	0.90 (0.76, 1.07)	0.81 (0.68, 0.97)	**0.83** **(0.69, 1.00)**	0.0102	**0.0808**
**Proximal colon cancer**								
EDIH	Absolute risk/cases	39/141	38/160	46/204	50/155	45/168	6	
EDIH	Relative risk	1 (reference)	1.16 (0.92, 1.47)	1.55 (1.23, 1.95)	1.19 (0.93, 1.52)	**1.41** **(1.10, 1.81)**	0.0134	**0.0808**
EDIP	Absolute risk/cases	42/147	41/185	47/178	44/160	44/158	2	
EDIP	Relative risk	1 (reference)	1.36 (1.07, 1.73)	1.34 (1.03, 1.73)	1.29 (0.98, 1.71)	**1.42** **(1.05, 1.92)**	0.0587	0.1966
HEI-2015	Absolute risk/cases	47/173	46/176	44/169	41/151	40/159	−7	
HEI-2015	Relative risk	1 (reference)	0.96 (0.78, 1.19)	0.92 (0.74, 1.14)	0.78 (0.62, 0.98)	**0.80** **(0.64, 1.01)**	0.0156	**0.0808**
**Distal colon and rectal cancer**								
EDIH	Absolute risk/cases	33/127	34/116	30/132	31/119	36/133	3	
EDIH	Relative risk	1 (reference)	0.91 (0.70, 1.19)	0.97 (0.74, 1.27)	0.95 (0.72, 1.25)	1.09 (0.82, 1.44)	0.4874	0.6174
EDIP	Absolute risk/cases	33/141	34/109	27/119	32/131	39/127	6	
EDIP	Relative risk	1 (reference)	0.80 (0.60, 1.05)	0.87 (0.65, 1.16)	1.02 (0.75, 1.39)	1.03 (0.74, 1.44)	0.5117	0.6375
HEI-2015	Absolute risk/cases	39/136	30/138	33/114	31/117	31/122	−8	
HEI-2015	Relative risk	1 (reference)	1.02 (0.80, 1.31)	0.88 (0.68, 1.14)	0.86 (0.66, 1.12)	0.92 (0.70, 1.20)	0.2774	0.4679
**Intestinal polyps**								
EDIH	Absolute risk/cases	551/1991	571/2030	594/2134	635/2211	649/2308	98	
EDIH	Relative risk	1 (reference)	1.05 (0.99, 1.12)	1.14 (1.07, 1.22)	1.18 (1.11, 1.27)	**1.23** **(1.15, 1.32)**	<0.0001	**0.0038**
EDIP	Absolute risk/cases	566/2143	604/2144	574/2113	616/2081	639/2193	73	
EDIP	Relative risk	1 (reference)	1.05 (0.98, 1.12)	1.07 (1.00, 1.15)	1.08 (1.01, 1.17)	**1.16** **(1.07, 1.26)**	0.0004	**0.0101**
HEI-2015	Absolute risk/cases	651/2307	626/2223	586/2172	585/2075	553/1896	−98	
HEI-2015	Relative risk	1 (reference)	0.97 (0.91, 1.03)	0.95 (0.89, 1.00)	0.89 (0.84, 0.95)	**0.84** **(0.78, 0.89)**	<0.0001	**0.0038**
**Invasive Breast cancer**								
EDIH	Absolute risk/cases	275/899	259/921	272/827	276/843	294/903	19	
EDIH	Relative risk	1 (reference)	1.09 (0.99, 1.20)	1.02 (0.92, 1.12)	1.06 (0.96, 1.18)	**1.19** **(1.07, 1.32)**	0.0032	**0.0405**
EDIP	Absolute risk/cases	264/926	266/936	285/874	285/866	275/791	11	
EDIP	Relative risk	1 (reference)	1.09 (0.99, 1.20)	1.07 (0.96, 1.19)	1.13 (1.01, 1.27)	**1.12** **(1.00, 1.27)**	0.0561	0.1966
HEI-2015	Absolute risk/cases	281/872	274/867	262/847	278/869	280/938	−1	
HEI-2015	Relative risk	1 (reference)	0.95 (0.86, 1.04)	0.90 (0.82, 1.00)	0.92 (0.83, 1.01)	0.99 (0.90, 1.09)	0.6135	0.6842
**ER+**								
EDIH	Absolute risk/cases	226/745	207/747	217/685	231/688	232/714	6	
EDIH	Relative risk	1 (reference)	1.08 (0.97, 1.20)	1.04 (0.93, 1.16)	1.07 (0.96, 1.20)	**1.16** **(1.04, 1.30)**	0.017	**0.0808**
EDIP	Absolute risk/cases	217/769	213/768	232/726	228/704	222/612	5	
EDIP	Relative risk	1 (reference)	1.09 (0.98, 1.22)	1.09 (0.97, 1.23)	1.13 (1.00, 1.29)	1.08 (0.94, 1.24)	0.1966	0.3735
HEI-2015	Absolute risk/cases	223/694	220/692	213/695	228/720	229/778	6	
HEI-2015	Relative risk	1 (reference)	0.95 (0.85, 1.05)	0.93 (0.84, 1.04)	0.95 (0.86, 1.06)	1.03 (0.92, 1.14)	0.6815	0.7117
**ER-**								
EDIH	Absolute risk/cases	29/100	37/123	36/105	29/100	38/127	9	
EDIH	Relative risk	1 (reference)	1.22 (0.93, 1.60)	1.05 (0.79, 1.40)	1.01 (0.75, 1.36)	1.31 (0.98, 1.76)	0.1847	0.3640
EDIP	Absolute risk/cases	31/108	35/104	35/106	37/124	31/113	0	
EDIP	Relative risk	1 (reference)	0.98 (0.73, 1.31)	1.02 (0.75, 1.39)	1.23 (0.89, 1.70)	1.19 (0.84, 1.68)	0.1796	0.3640
HEI-2015	Absolute risk/cases	38/123	31/110	31/107	33/104	35/111	−3	
HEI-2015	Relative risk	1 (reference)	0.87 (0.67, 1.12)	0.83 (0.64, 1.08)	0.79 (0.61, 1.04)	0.85 (0.65, 1.12)	0.1725	0.3640
**PR+**								
EDIH	Absolute risk/cases	183/626	177/636	188/588	195/575	194/604	11	
EDIH	Relative risk	1 (reference)	1.10 (0.99, 1.24)	1.08 (0.96, 1.21)	1.09 (0.96, 1.23)	**1.20** **(1.06, 1.36)**	0.0085	**0.0808**
EDIP	Absolute risk/cases	184/648	177/649	197/635	195/597	184/500	0	
EDIP	Relative risk	1 (reference)	1.12 (0.99, 1.26)	1.17 (1.03, 1.33)	1.18 (1.03, 1.36)	1.09 (0.94, 1.27)	0.1438	0.3277
HEI-2015	Absolute risk/cases	190/597	187/580	181/593	191/615	189/644	−1	
HEI-2015	Relative risk	1 (reference)	0.92 (0.82, 1.04)	0.92 (0.82, 1.03)	0.94 (0.84, 1.06)	0.98 (0.87, 1.10)	0757	0.8946
**PR-**								
EDIH	Absolute risk/cases	66/206	62/222	62/195	60/204	70/225	4	
EDIH	Relative risk	1 (reference)	1.08 (0.89, 1.31)	0.96 (0.78, 1.19)	1.02 (0.83, 1.26)	1.16 (0.94, 1.43)	0.2408	0.4679
EDIP	Absolute risk/cases	61/220	65/214	65/182	66/222	63/214	2	
EDIP	Relative risk	1 (reference)	0.96 (0.79, 1.18)	0.83 (0.66, 1.03)	1.04 (0.83, 1.31)	1.06 (0.83, 1.36)	0.5338	0.6579
HEI-2015	Absolute risk/cases	68/212	59/212	60/204	65/199	69/225	1	
HEI-2015	Relative risk	1 (reference)	0.96 (0.79, 1.16)	0.91 (0.74, 1.10)	0.88 (0.72, 1.07)	0.99 (0.81, 1.21)	0.6136	0.7117
**HER2+**								
EDIH	Absolute risk/cases	27/86	24/85	26/86	28/85	30/100	3	
EDIH	Relative risk	1 (reference)	0.99 (0.73, 1.35)	1.03 (0.75, 1.41)	1.02 (0.74, 1.42)	1.24 (0.90, 1.71)	0.1823	0.3640
EDIP	Absolute risk/cases	25/81	26/97	27/91	31/86	26/87	1	
EDIP	Relative risk	1 (reference)	1.25 (0.91, 1.72)	1.21 (0.86, 1.71)	1.20 (0.83, 1.74)	1.29 (0.86, 1.92)	0.3012	0.4766
HEI-2015	Absolute risk/cases	28/97	26/84	25/84	28/90	28/87	0	
HEI-2015	Relative risk	1 (reference)	0.84 (0.63, 1.13)	0.84 (0.63, 1.14)	0.91 (0.68, 1.22)	0.89 (0.66, 1.22)	0.5863	0.6749
**HER2-**								
EDIH	Absolute risk/cases	208/698	198/708	205/639	207/647	216/660	8	
EDIH	Relative risk	1 (reference)	1.09 (0.98, 1.22)	1.03 (0.92, 1.16)	1.08 (0.96, 1.21)	**1.15** **(1.02, 1.29)**	0.0424	0.1696
EDIP	Absolute risk/cases	203/726	200/722	214/665	210/660	207/579	4	
EDIP	Relative risk	1 (reference)	1.09 (0.98, 1.22)	1.06 (0.94, 1.20)	1.13 (0.99, 1.29)	1.09 (0.95, 1.26)	0.1868	0.3640
HEI-2015	Absolute risk/cases	212/654	202/652	198/662	208/657	214/727	2	
HEI-2015	Relative risk	1 (reference)	0.95 (0.85, 1.06)	0.94 (0.84, 1.05)	0.92 (0.82, 1.03)	1.01 (0.90, 1.13)	0.9486	0.9486
**ER- PR- HER2+**								
EDIH	Absolute risk/cases	7/25	8/23	8/25	7/25	7/22	0	
EDIH	Relative risk	1 (reference)	0.92 (0.51, 1.65)	1.01 (0.56, 1.82)	1.01 (0.55, 1.84)	0.89 (0.47, 1.68)	0.8008	0.8008
EDIP	Absolute risk/cases	6/19	8/31	8/21	9/25	5/24	−1	
EDIP	Relative risk	1 (reference)	1.88 (1.00, 3.51)	1.35 (0.66, 2.77)	1.71 (0.81, 3.58)	1.73 (0.78, 3.83)	0.2972	0.4766
HEI-2015	Absolute risk/cases	9/30	5/22	7/23	7/23	8/22	−1	
HEI-2015	Relative risk	1 (reference)	0.72 (0.42, 1.26)	0.75 (0.43, 1.31)	0.76 (0.44, 1.34)	0.76 (0.42, 1.36)	0.3750	0.5182
**Luminal A**								
EDIH	Absolute risk/cases	189/640	173/625	183/575	188/582	190/576	1	
EDIH	Relative risk	1 (reference)	1.06 (0.95, 1.19)	1.03 (0.91, 1.16)	1.08 (0.95, 1.22)	**1.11** **(0.98, 1.26)**	0.1054	0.2762
EDIP	Absolute risk/cases	182/660	178/655	190/599	185/577	188/507	6	
EDIP	Relative risk	1 (reference)	1.10 (0.98, 1.24)	1.07 (0.94, 1.22)	1.11 (0.97, 1.28)	1.08 (0.93, 1.25)	0.3097	0.4766
HEI-2015	Absolute risk/cases	187/579	181/582	179/593	186/593	190/651	3	
HEI-2015	Relative risk	1 (reference)	0.95 (0.85, 1.07)	0.95 (0.84, 1.06)	0.93 (0.83, 1.05)	1.02 (0.90, 1.14)	0.9289	0.9289
**Luminal B**								
EDIH	Absolute risk/cases	19/59	16/61	18/60	20/59	22/76	3	
EDIH	Relative risk	1 (reference)	1.05 (0.72, 1.51)	1.06 (0.72, 1.54)	1.05 (0.71, 1.55)	1.40 (0.96, 2.04)	0.0885	0.2491
EDIP	Absolute risk/cases	19/61	17/65	20/69	21/59	19/61	0	
EDIP	Relative risk	1 (reference)	1.07 (0.73, 1.56)	1.15 (0.77, 1.71)	1.02 (0.66, 1.58)	1.11 (0.70, 1.77)	0.7315	0.7315
HEI-2015	Absolute risk/cases	19/65	19/59	18/60	20/67	20/64	1	
HEI-2015	Relative risk	1 (reference)	0.88 (0.62, 1.26)	0.90 (0.63, 1.29)	1.01 (0.71, 1.44)	0.98 (0.68, 1.41)	0.9019	0.9019
**Triple negative**								
EDIH	Absolute risk/cases	18/57	24/83	21/61	17/63	25/84	7	
EDIH	Relative risk	1 (reference)	1.43 (1.01, 2.01)	1.05 (0.72, 1.54)	1.09 (0.74, 1.60)	**1.49** **(1.02, 2.16)**	0.1386	0.3277
EDIP	Absolute risk/cases	19/66	21/65	22/65	24/81	18/71	−1	
EDIP	Relative risk	1 (reference)	0.97 (0.67, 1.40)	0.98 (0.66, 1.45)	1.24 (0.82, 1.85)	1.13 (0.73, 1.76)	0.3704	0.5182
HEI-2015	Absolute risk/cases	24/75	21/70	19/69	19/61	23/73	−1	
HEI-2015	Relative risk	1 (reference)	0.91 (0.66, 1.27)	0.89 (0.64, 1.24)	0.78 (0.55, 1.10)	0.93 (0.66, 1.31)	0.4526	0.5950
**Invasive ductal carcinoma**								
EDIH	Absolute risk/cases	144/463	140/482	143/474	153/478	158/500	14	
EDIH	Relative risk	1 (reference)	1.07 (0.94, 1.22)	1.08 (0.94, 1.24)	1.11 (0.97, 1.28)	**1.20** **(1.04, 1.38)**	0.0119	0.0808
EDIP	Absolute risk/cases	146/489	143/512	152/465	153/492	146/439	0	
EDIP	Relative risk	1 (reference)	1.08 (0.94, 1.23)	1.00 (0.87, 1.16)	1.11 (0.95, 1.30)	1.06 (0.89, 1.25)	0.4745	0.6112
HEI-2015	Absolute risk/cases	149/459	140/485	142/450	154/506	154/497	6	
HEI-2015	Relative risk	1 (reference)	1.02 (0.89, 1.16)	0.93 (0.82, 1.06)	1.05 (0.92, 1.19)	1.04 (0.91, 1.19)	0.5326	0.6529
**Invasive lobular carcinoma**								
EDIH	Absolute risk/cases	25/87	20/81	23/65	21/68	28/81	3	
EDIH	Relative risk	1 (reference)	1.04 (0.76, 1.42)	0.88 (0.63, 1.24)	0.97 (0.69, 1.37)	1.23 (0.87, 1.72)	0.3275	0.4787
EDIP	Absolute risk/cases	22/96	23/62	24/89	23/64	24/71	2	
EDIP	Relative risk	1 (reference)	0.70 (0.49, 0.98)	1.05 (0.76, 1.47)	0.81 (0.55, 1.18)	1.00 (0.67, 1.48)	0.8900	0.8900
HEI-2015	Absolute risk/cases	24/80	24/57	20/77	24/82	23/86	−1	
HEI-2015	Relative risk	1 (reference)	0.66 (0.47, 0.93)	0.87 (0.63, 1.19)	0.89 (0.65, 1.23)	0.91 (0.66, 1.26)	0.9513	0.9513
**Localized**								
EDIH	Absolute risk/cases	186/622	179/633	185/580	190/582	200/615	14	
EDIH	Relative risk	1 (reference)	1.08 (0.97, 1.21)	1.03 (0.92, 1.16)	1.07 (0.94, 1.21)	**1.18** **(1.04, 1.34)**	0.0193	**0.0863**
EDIP	Absolute risk/cases	176/630	190/658	193/597	196/606	185/541	9	
EDIP	Relative risk	1 (reference)	1.13 (1.01, 1.28)	1.08 (0.95, 1.23)	1.17 (1.02, 1.34)	1.14 (0.98, 1.32)	0.0836	0.2444
HEI-2015	Absolute risk/cases	189/584	192/595	179/597	188/598	192/658	3	
HEI-2015	Relative risk	1 (reference)	0.97 (0.86, 1.09)	0.95 (0.84, 1.07)	0.94 (0.83, 1.06)	1.03 (0.91, 1.16)	0.8605	0.8605
**Regional/distant**								
EDIH	Absolute risk/cases	67/219	62/230	67/199	66/215	70/229	3	
EDIH	Relative risk	1 (reference)	1.09 (0.90, 1.32)	0.98 (0.80, 1.21)	1.09 (0.89, 1.34)	1.20 (0.98, 1.48)	0.1030	0.2762
EDIP	Absolute risk/cases	69/234	59/220	71/224	67/219	67/195	−2	
EDIP	Relative risk	1 (reference)	1.00 (0.82, 1.22)	1.07 (0.86, 1.32)	1.11 (0.88, 1.39)	1.07 (0.83, 1.37)	0.4475	0.5950
HEI-2015	Absolute risk/cases	71/226	60/216	66/201	67/220	69/229	−2	
HEI-2015	Relative risk	1 (reference)	0.92 (0.77, 1.12)	0.85 (0.70, 1.03)	0.92 (0.76, 1.12)	0.97 (0.80, 1.18)	0.7019	0.7117
**Endometrial cancer**								
EDIH	Absolute risk/cases	46/74	47/88	53/79	52/73	62/89	16	
EDIH	Relative risk	1 (reference)	1.30 (0.94, 1.79)	1.24 (0.89, 1.74)	1.24 (0.88, 1.76)	**1.63** **(1.16, 2.30)**	0.0110	**0.0808**
EDIP	Absolute risk/cases	50/81	49/93	49/67	55/84	58/78	8	
EDIP	Relative risk	1 (reference)	1.21 (0.88, 1.68)	0.94 (0.65, 1.36)	1.27 (0.87, 1.85)	1.39 (0.92, 2.09)	0.1466	0.3277
HEI-2015	Absolute risk/cases	56/67	52/84	57/98	42/81	53/73	−3	
HEI-2015	Relative risk	1 (reference)	1.16 (0.84, 1.60)	1.27 (0.93, 1.74)	1.01 (0.72, 1.41)	0.88 (0.62, 1.24)	0.3151	0.4766
**Endometroid**								
EDIH	Absolute risk/cases	32/49	22/50	35/53	38/52	42/62	10	
EDIH	Relative risk	1 (reference)	1.14 (0.76, 1.71)	1.30 (0.86, 1.96)	1.38 (0.90, 2.10)	**1.74** **(1.15, 2.64)**	0.0058	**0.0630**
EDIP	Absolute risk/cases	28/54	32/59	32/42	33/55	44/56	16	
EDIP	Relative risk	1 (reference)	1.19 (0.80, 1.77)	0.90 (0.57, 1.43)	1.28 (0.80, 2.03)	1.52 (0.93, 2.50)	0.117	0.2964
HEI-2015	Absolute risk/cases	36/48	29/53	45/66	34/54	25/45	−11	
HEI-2015	Relative risk	1 (reference)	1.03 (0.69, 1.52)	1.21 (0.83, 1.77)	0.94 (0.63, 1.41)	0.77 (0.50, 1.18)	0.2181	0.4043
**Non-endometroid**								
EDIH	Absolute risk/cases	15/25	19/39	19/25	17/21	16/27	1	
EDIH	Relative risk	1 (reference)	1.57 (0.94, 2.64)	1.07 (0.60, 1.92)	0.97 (0.52, 1.80)	1.36 (0.75, 2.48)	0.7359	0.7359
EDIP	Absolute risk/cases	20/27	13/33	19/26	18/29	16/22	−4	
EDIP	Relative risk	1 (reference)	1.21 (0.70, 2.09)	1.01 (0.54, 1.88)	1.18 (0.62, 2.27)	1.03 (0.50, 2.13)	0.9443	0.9443
HEI-2015	Absolute risk/cases	15/19	20/31	15/31	15/27	21/29	6	
HEI-2015	Relative risk	1 (reference)	1.52 (0.86, 2.70)	1.43 (0.80, 2.55)	1.21 (0.66, 2.20)	1.25 (0.68, 2.28)	0.7907	0.7907
**Ovarian cancer**								
EDIH	Absolute rate/cases	35/60	34/52	25/54	36/41	37/53	2	
EDIH	Relative risk	1 (reference)	0.95 (0.65, 1.39)	1.04 (0.70, 1.53)	0.85 (0.55, 1.30)	1.16 (0.77, 1.75)	0.5950	0.6749
EDIP	Absolute rate/cases	37/64	31/63	32/44	33/46	34/43	−3	
EDIP	Relative risk	1 (reference)	1.02 (0.70, 1.49)	0.76 (0.49, 1.18)	0.85 (0.54, 1.36)	0.89 (0.54, 1.47)	0.4541	0.5950
HEI-2015	Absolute risk/cases	33/47	28/45	36/42	34/69	37/57	4	
HEI-2015	Relative risk	1 (reference)	0.93 (0.62, 1.40)	0.84 (0.55, 1.29)	1.37 (0.93, 2.01)	1.09 (0.73, 1.64)	0.2412	0.4365
**Serous**								
EDIH	Absolute risk/cases	14/27	16/25	9/23	17/15	17/24	3	
EDIH	Relative risk	1 (reference)	1.06 (0.60, 1.85)	1.06 (0.59, 1.91)	0.75 (0.38, 1.47)	1.26 (0.68, 2.32)	0.7023	0.7117
EDIP	Absolute risk/cases	18/28	12/35	14/20	15/10	14/21	−4	
EDIP	Relative risk	1 (reference)	1.38 (0.80,2.39)	0.88 (0.46, 1.71)	0.48 (0.21, 1.11)	1.41 (0.54, 2.42)	0.5589	0.6637
HEI-2015	Absolute risk/cases	16/23	11/19	15/21	16/28	14/23	−2	
HEI-2015	Relative risk	1 (reference)	0.80 (0.44, 1.48)	0.86 (0.47, 1.57)	1.13 (0.64, 2.00)	0.90 (0.49, 1.66)	0.9040	0.9040
**Non-Serous**								
EDIH	Absolute risk/cases	20/33	18/27	16/31	18/26	20/29	0	
EDIH	Relative risk/cases	1 (reference)	0.86 (0.51, 1.44)	1.01 (0.60, 1.70)	0.92 (0.53, 1.60)	1.10 (0.63, 1.91)	0.6922	0.7117
EDIP	Absolute risk/cases	19/36	18/28	18/24	18/36	20/22	1	
EDIP	Relative risk	1 (reference)	0.75 (0.44, 1.28)	0.66 (0.37, 1.91)	1.04 (0.58, 1.87)	0.70 (0.36, 1.39)	0.5761	0.6736
HEI-2015	Absolute risk/cases	16/23	15/27	20/21	19/41	22/34	−6	
HEI-2015	Relative risk	1 (reference)	1.14 (0.65, 1.99)	0.87 (0.48, 1.58)	1.66 (0.98, 2.80)	1.32 (0.76, 2.31)	0.1355	0.3277
**Lung cancer**								
EDIH	Absolute risk/cases	116/482	108/408	107/410	113/417	119/399	3	
EDIH	Relative risk	1 (reference)	0.94 (0.82, 1.08)	0.97 (0.84, 1.11)	1.01 (0.88, 1.16)	0.89 (0.77, 1.03)	0.2609	0.4507
EDIP	Absolute risk/cases	110/547	109/464	114/397	117/377	113/331	3	
EDIP	Relative risk	1 (reference)	1.05 (0.92, 1.20)	0.97 (0.84, 1.13)	1.03 (0.88, 1.21)	1.02 (0.85, 1.21)	0.9331	0.9331
HEI-2015	Absolute risk/cases	130/520	112/439	110/415	113/387	99/354	−31	
HEI-2015	Relative risk	1 (reference)	0.97 (0.86, 1.11)	0.99 (0.87, 1.13)	0.97 (0.84, 1.11)	0.91 (0.78, 1.05)	0.2555	0.4507
**Small cell**								
EDIH	Absolute risk/cases	7/25	8/32	7/26	6/29	9/26	2	
EDIH	Relative risk	1 (reference)	1.43 (0.83, 2.45)	1.02 (0.57, 1.83)	1.15 (0.64, 2.04)	0.81 (0.44, 1.48)	0.3198	0.4766
EDIP	Absolute risk/cases	8/31	5/38	6/23	10/28	8/18	0	
EDIP	Relative risk	1 (reference)	1.55 (0.92, 2.61)	0.93 (0.50, 1.72)	1.28 (0.68, 2.42)	0.83 (0.40, 1.74)	0.6212	0.6842
HEI-2015	Absolute risk/cases	7/33	7/27	7/32	8/23	7/23	0	
HEI-2015	Relative risk	1 (reference)	1.21 (0.72, 2.04)	1.81 (1.09, 3.02)	1.51 (0.86, 2.65)	**1.79** **(1.00, 3.23)**	0.028	0.1182
**Non-small cell**								
EDIH	Absolute risk/cases	57/249	59/183	51/204	55/225	59/198	2	
EDIH	Relative risk	1 (reference)	0.81 (0.67, 0.99)	0.94 (0.77, 1.14)	1.06 (0.88, 1.30)	0.89 (0.72, 1.09)	0.7986	0.7986
EDIP	Absolute risk/cases	57/275	50/218	61/205	59/192	54/169	−3	
EDIP	Relative risk	1 (reference)	0.98 (0.80, 1.18)	1.00 (0.81, 1.23)	1.04 (0.83, 1.31)	1.04 (0.81, 1.34)	0.6585	0.7117
HEI-2015	Absolute risk/cases	64/251	56/219	53/207	57/197	51/184	−13	
HEI-2015	Relative risk	1 (reference)	0.98 (0.81, 1.18)	0.97 (0.80, 1.17)	0.95 (0.78, 1.16)	0.90 (0.73, 1.11)	0.3434	0.4924

^a^ Values presented are hazard ratios (HR) and 95% confidence intervals (95% CI) for relative risk and incidence rate per 100,000 person years for absolute risk. HRs were derived from multivariable-adjusted Cox proportional hazards regression models adjusted for the following baseline covariates: age at enrollment, physical activity, race and ethnicity, educational level, family history of cancer, number of hormones used, comorbidity score, baseline cardiovascular disease status, baseline lung disease, number of supplements used, non-steroidal anti-inflammatory drug use, hormone therapy study arm, baseline hormone therapy ever, oral contraceptive duration, pack years of smoking, coffee/tea and total alcohol intake. Colorectal cancer and subtype analyses were additionally adjusted for colorectal cancer screening. Invasive breast cancer and subtype analyses were additionally adjusted for months of breast-feeding, age at menopause, mammogram ever, parity, bilateral oophorectomy, passive smoking and Gail 5-year risk score. Endometrial cancer and ovarian cancer analyses were additionally adjusted for age at first birth, age at menarche, age at menopause, months of breast-feeding and parity. Ovarian cancer analyses were further adjusted for tubal ligation. Lung cancer analyses were additionally adjusted for smoking status and passive smoking. ^b^ Details of cancer site definition are included in Appendix A. ^c^ Each dietary pattern was adjusted for the same covariates using the residual method prior to estimating the incidence rates. Incidence rate per 100,000 person years were calculated using the number of cases of each cancer within a quintile of the dietary score divided by the sum of year-to-event within that dietary score quintile and then multiplied by 100,000. The Q5-Q1 incidence rate per 100,000 person years was calculated using the incidence rate per 100,000 person-year of the 5th quintile of a dietary score minus that of the 1st quintile of the dietary score. ^d^ The *p* value for linear trend was estimated in the same multivariable-adjusted models by assigning the quintile-specific median value of each dietary pattern to all participants in the quintile and modelling as an ordinal variable. The *p* value for linear trend was adjusted for false discovery rate using the Benjamini and Hotchberg approach. ^e^ EDIH, empirical dietary index for hyperinsulinemia score assesses the ability of the dietary pattern to contribute to insulin hypersecretion—higher EDIH scores reflect more hyperinsulinemic dietary patterns; EDIP, empirical dietary inflammatory pattern score assesses the ability of the dietary pattern to contribute to chronic systemic inflammation—higher EDIP scores reflect more pro-inflammatory dietary patterns; HEI-2015, healthy eating index-2015 assesses adherence to the 2015–2020 Dietary Guidelines for Americans—higher HEI-2015 scores are indicative of greater adherence and higher dietary quality. EDIH and EDIP are positively correlated, whereas both scores are inversely correlated with HEI-2015, i.e., more hyperinsulinemic or pro-inflammatory dietary patterns are of lower dietary quality.

**Figure 2 cancers-15-01756-f002:**
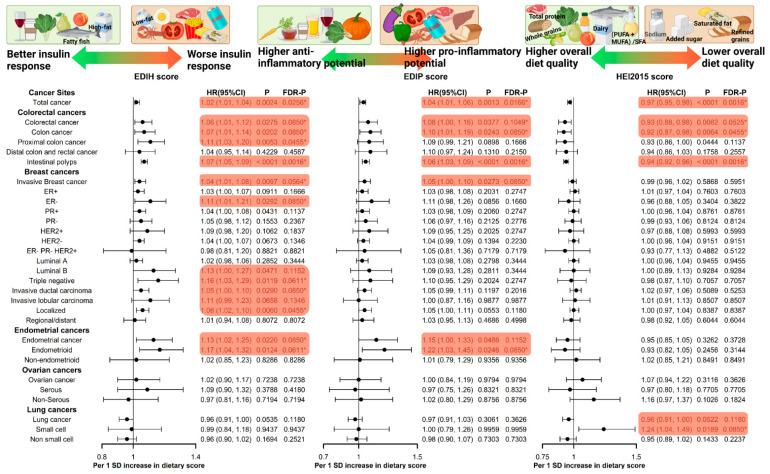
Associations between a 1 standard deviation increment in hyperinsulinemic, pro-inflammatory dietary patterns or higher overall dietary quality, and risk of incident total cancer, site-specific cancers, and pathological subtypes. Values are hazard ratios (HR) and 95% confidence intervals (95% CI) for cancer risk per 1 standard deviation increment in dietary score. Values in colored background/asterisk represent elevated risk (EDIH/EDIP), reduced risk (HEI-2015), or significant FDR-adjusted *p*-value at <0.15. HRs were derived from multivariable-adjusted Cox proportional hazards regression models adjusted for the following baseline covariates: age at enrollment, physical activity, race and ethnicity, educational level, family history of cancer, number of hormones used, comorbidity score, baseline cardiovascular disease status, baseline lung disease, number of supplements used, non-steroidal anti-inflammatory drug use, hormone therapy study arm, baseline hormone therapy ever, oral contraceptive duration, pack years of smoking, coffee/tea, and total alcohol intake. Colorectal cancer and subtype analyses were additionally adjusted for colorectal cancer screening. Invasive breast cancer and subtype analyses were additionally adjusted for months of breast-feeding, age at menopause, mammogram ever, parity, bilateral oophorectomy, passive smoking, Gail 5-year risk score. Endometrial cancer and ovarian cancer analyses were additionally adjusted for age at first birth, age at menarche, age at menopause, months of breast-feeding, and parity.

### 3.5. EDIP and Cancer Risk (Table 2, Figure 2)

Women in the highest EDIP quintile were at greater risk of total cancer compared to those in the lowest quintile, with a multivariable-adjusted incidence rate difference of 41 per 100,000 person years and corresponding HR (95% CI) of 1.08 (1.02–1.15) (Table 2). A 1 sd increment in EDIP was associated with higher risk of colorectal cancer, colon cancer and marginally with proximal and distal colon or rectal cancer. EDIP was also associated with intestinal polyps. Higher EDIP was associated with greater risk of overall breast cancer and ER-negative cancer. Furthermore, higher EDIP was associated with greater risk for endometrial cancer, but not ovarian cancer or lung cancer (Figure 2). The EDIP metabolomics score was generally not associated with cancer risk (Table 3).

### 3.6. HEI-2015 and Cancer Risk (Table 2, Figure 2)

Women classified in the highest HEI-2015 quintile were at lower risk of total cancer compared to those in the lowest quintile, with a multivariable-adjusted incidence rate difference of −49 per 100,000 person years and corresponding HR (95% CI) of 0.93 (0.89–0.98) (Table 2). Higher HEI-2015 was associated with lower risk of colorectal cancer, colon cancer and proximal colon cancer but not distal colon or rectal cancer. HEI-2015 was also strongly associated with intestinal polyps. Unlike EDIH and EDIP, HEI-2015 was not associated with overall breast cancer or its subtypes nor with endometrial or ovarian cancers but was marginally inversely associated with overall lung cancer, though positively associated with small-cell lung cancer. Higher HEI-2015 metabolomics profile score was associated with lower risk for overall lung cancer, HR, 0.46 (0.24–0.90) (Table 3).

### 3.7. Sensitivity Analyses and Subgroup Analyses (Appendix A)

Additional adjustment for BMI and type 2 diabetes, major mediators strongly associated with EDIH and EDIP in previous studies, did not materially change the results, though results for endometrial cancer were attenuated and no longer statistically significant (Appendix A). Findings from subgroup analyses are reported in Appendix A. *p* values for the interactions of the dietary patterns and the potential effect modifiers were generally not significant. The results for total cancer, breast cancer and endometrial cancer remained robust for EDIH after mutually adjusting for EDIP and HEI-2015 (Appendix A).

## 4. Discussion

### 4.1. Principal Findings, Strengths and Weaknesses in Relation to Other Studies

The present study employed two empirical hypothesis-oriented dietary indices to investigate associations between diets that contribute to chronic hyperinsulinemia (EDIH) or chronic systemic inflammation (EDIP) and risk of developing total cancer and site-specific cancers among postmenopausal women, while also examining these associations with an established index of overall dietary quality (HEI-2015). While all three dietary indices were significantly associated with risk of total cancer, colorectal cancer and intestinal polyps, EDIH and EDIP were further associated with risk of breast cancer and endometrial cancer. In addition, EDIH was associated with multiple breast cancer subtypes including the more aggressive triple negative breast cancer.

Although a previous study found that higher EDIH was associated with higher total cancer mortality in both men and women [32], most previous studies have examined single cancer sites and reported similar findings to ours. For example, in the NHS (another all-female cohort), EDIH and EDIP were associated with higher risk of colorectal cancer and its anatomic subsites except the rectum [6,8]. EDIH was also associated with higher risk of digestive system cancers and accessory organs [7]. In the WHI, dietary inflammatory potential assessed using a literature-derived nutrient-based dietary inflammatory index (DII) was associated with higher risk of colorectal cancer [33] similar to the current study. Given that DII was calculated with total intake (nutrients plus supplements), findings may not be directly comparable to EDIP, which is exclusively food-based. Though different polyp types were not adjudicated in WHI, the strong associations of EDIH and EDIP with intestinal polyps warrant future studies to test if dietary intakes may inform risk stratification in colorectal cancer screening for improved preventive strategies. Associations between EDIH and EDIP with total cancer and colorectal cancer were consistent with findings for HEI-2015. A study in WHI also found that higher HEI-2010 score was associated with lower colorectal cancer risk [34].

Although EDIH and EDIP were associated with higher risk of overall breast cancer and ER-negative subtype, only EDIH was associated with risk of multiple other breast cancer subtypes, while HEI-2015 showed no association. Only one study has examined EDIH in relation to breast cancer risk, and found higher risk for overall breast cancer with stronger associations for ER- and HER2+ tumors [35]. Two studies in the WHI found no associations between the DII and risk of overall breast cancer/subtypes [36,37]. The HEI and most other dietary indices have not been consistently associated with breast cancer risk [38]. The EDIH and EDIP dietary patterns are more strongly related to obesity and type 2 diabetes than most traditional dietary patterns [24,30,31,39]. Obesity and diabetes drive risk of the same cancers including colorectum, postmenopausal breast, endometrium and ovary, among other sites [3,40]. In the current study, both EDIH and EDIP were strongly associated with endometrial cancer risk, which attenuated after adjusting for obesity as a mediator. Similarly, a study in NHS and NHS-II cohorts applied the EDIH and EDIP scores and found strong associations with endometrial cancer, and showed that BMI mediated 84% and 93% of the associations, respectively [9]. Obesity is linked to inflammation and insulin resistance [41], and a study that analyzed data on 1.2 million women, found that each 10-kg/m^2^ increment in BMI was associated with a nearly 3-fold increase in endometrial cancer risk [42]. While EDIH has not been studied in relation to ovarian cancer risk, the null association with EDIP was consistent with a study in NHS and NHS-II [43].

The metabolomic profile of HEI-2015 had 23 metabolites that overlapped with the EDIH, and had 17 metabolites overlapped with the EDIP, which may partly explain the higher correlation between the HEI-2015 related metabolomics score is and EDIH-related metabolomics score, compared with the EDIP-related metabolomics score. We observed no associations between the metabolomics profile scores of the three dietary patterns and cancer risk, except for the associations between HEI-2015 score and overall lung cancer. We had 441 cancer cases in the metabolomics sample compared to 18,768 in the overall sample. It is therefore possible that the lack of associations in contrast to the dietary analysis is indicative of the low statistical power in our metabolomics sample. The association observed had wide 95% CIs, reflecting potentially unstable point estimates. Nevertheless, we characterized the correlations of metabolomics classes with the food group components of the dietary scores, which yielded novel and confirmatory findings. In the two previous metabomomics studies [16,17], higher levels of nine CEs, one glycerophosphoserine, trigonelline and eicosapentanoate were associated with lower EDIH score [16]. In the current study, EDIH showed inverse associations with three CEs and one glycerophosphoserine, though using a different method to derive metabolomic profile scores. Higher CE levels were associated with higher intake of wine and fruit and with lower intake of red meat, sugar-sweetened beverages, and processed meat. It is therefore possible that a low EDIH diet may reduce disease risks via CE’s greater efficiency in clearing blood remnants of lipid metabolism [16,44]. We found that higher plasma levels of alkaloids and purines were associated with lower EDIH/EDIP and with higher coffee/tea which contribute to lower EDIH/EDIP scores. Laboratory studies suggest that specific alkaloids can intervene in the insulin signal transduction pathway, and reverse molecular defects that could otherwise lead to insulin resistance and glucose intolerance [45]. These alkaloids may be involved in pathophysiological processes associated with insulin resistance, β-cell failure, oxidative stress and inflammation [45].

### 4.2. Strengths and Weaknesses of the Study

Our study has several strengths. We investigated EDIH, EDIP and HEI-2015 in relation to multiple cancers, and dietary pattern-related metabolomics profiles and their association with cancer risk. We estimated multivariable-adjusted incidence rates in addition to the usual relative risk estimates, providing better clinical and public health context for interpreting the relative risk estimates, e.g., incidence rate among the non-exposed (reference—quintile 1). We adjusted *p*-values to minimize the potential for false discovery. Potential limitations include using self-reported dietary intake though the FFQ was evaluated for bias and precision [22]. Although the FFQ and metabolites were single measurements, previous studies have shown that diet in adults and plasma metabolites remain stable overtime [46,47]. Though the sample sizes for the cancer risk analyses were large, the number of cancer cases in the metabolomics sample was small, precluding robust associations. Additionally, though we adjusted for numerous potential confounding factors, residual confounding may persist [48,49].

### 4.3. Possible Implications and Conclusions

In summary, our findings suggest that hyperinsulinemic and pro-inflammatory dietary patterns, as well as overall dietary quality, are associated with risk for several cancers among postmenopausal women, supporting further investigation of these dietary patterns in relation to cancer risk in dietary intervention studies to modify cancer risk.

## Figures and Tables

**Table 3 cancers-15-01756-t003:** Dietary pattern-related metabolomics signatures in relation to total cancer and site-specific cancer risks ^a,b,c,d,e,f^.

	T1	T2	T3	P for Linear Trend ^c^	1 sd Increment	P for Continuous Dietary Score ^e^
Overall cancer/cases						
EDIH metabolomics score	165	133	143			
MV-adjusted	1	0.80 (0.63, 1.01)	0.93 (0.72, 1.20)	0.5171	0.99 (0.78, 1.26)	0.9115
EDIP metabolomics score	179	140	122			
MV-adjusted	1	0.83 (0.66, 1.04)	0.81 (0.62, 1.05)	0.0893	0.76 (0.58, 1.00)	0.0531
HEI metabolomics score	161	130	150			
MV-adjusted	1	0.82 (0.64, 1.05)	0.94 (0.73, 1.22)	0.6573	0.86 (0.68, 1.09)	0.2083
Colorectal cancer/cases						
EDIH metabolomics score	25	16	18			
MV-adjusted	1	0.34 (0.14, 0.85)	0.52 (0.21, 1.28)	0.1259	0.67 (0.27, 1.63)	0.3777
EDIP metabolomics score	28	13	18			
MV-adjusted	1	0.38 (0.15, 0.96)	0.63 (0.25, 1.61)	0.3514	0.57 (0.20, 1.60)	0.2845
HEI metabolomics score	21	13	25			
MV-adjusted	1	0.28 (0.10, 0.77)	1.56 (0.66, 3.71)	0.2690	1.33 (0.58, 3.07)	0.5007
Colon cancer/cases						
EDIH metabolomics score	19	10	17			
MV-adjusted	1	0.38 (0.14, 1.03)	0.74 (0.29, 1.89)	0.5184	1.04 (0.39, 2.73)	0.9413
EDIP metabolomics score	21	9	16			
MV-adjusted	1	0.41 (0.14, 1.22)	0.73 (0.26, 2.04)	0.6204	0.61 (0.20 1.92)	0.3987
HEI metabolomics score	19	10	17			
MV-adjusted	1	0.27 (0.09, 0.77)	0.94 (0.37, 2.40)	0.9655	0.78 (0.31, 1.96)	0.5954
Intestinal Polyps/cases						
EDIH metabolomics score	67	77	100			
MV-adjusted	1	0.96 (0.63, 1.44)	1.13 (0.72, 1.76)	0.5515	1.29 (0.84, 1.98)	0.2458
EDIP metabolomics score	61	75	108			
MV-adjusted	1	1.17 (0.75, 1.84)	1.46 (0.91, 2.34)	0.1136	1.51 (0.93, 2.46)	0.0989
HEI metabolomics score	95	80	69			
MV-adjusted	1	0.89 (0.60, 1.33)	0.72 (0.47, 1.11)	0.1421	0.76 (0.51, 1.13)	0.1701
Invasive breast cancer/cases						
EDIH metabolomics score	44	33	41			
MV-adjusted	1	0.74 (0.46, 1.18)	1.00 (0.62, 1.62)	0.9814	1.39 (0.87, 2.20)	0.1649
EDIP metabolomics score	41	37	40			
MV-adjusted	1	0.95 (0.60, 1.52)	1.16 (0.71, 1.89)	0.5685	1.01 (0.60, 1.71)	0.9649
HEI metabolomics score	38	39	41			
MV-adjusted	1	1.02 (0.64, 1.64)	1.03 (0.62, 1.69)	0.9158	1.00 (0.65, 1.56)	0.9812
Endometrial cancer/cases						
EDIH metabolomics score	5	5	12			
MV-adjusted	1	0.44 (0.10, 1.87)	1.78 (0.51, 6.27)	0.2407	3.58 (0.96, 13.29)	0.0566
EDIP metabolomics score	6	5	11			
MV-adjusted	1	0.70 (0.18, 2.68)	1.25 (0.36, 4.35)	0.6640	2.67 (0.63, 11.28)	0.1805
HEI metabolomics score	7	10	5			
MV-adjusted	1	1.23 (0.39, 3.93)	0.54 (0.13, 2.25)	0.3963	0.41 (0.12, 1.43)	0.1627
Lung cancer/cases						
EDIH metabolomics score	23	28	24			
MV-adjusted	1	1.25 (0.69, 2.27)	1.10 (0.56, 2.14)	0.7803	0.88 (0.47, 1.65)	0.6964
EDIP metabolomics score	33	39	13			
MV-adjusted	1	0.96 (0.56, 1.65)	0.52 (0.25, 1.12)	0.1340	0.57 (0.28, 1.14)	0.1115
HEI metabolomics score	41	18	16			
MV-adjusted	1	**0.49 (0.26, 0.92)**	0.49 (0.24, 1.01)	**0.0289**	**0.46 (0.24, 0.90)**	**0.0227**

^a^ We used metabolomics data from a matched case–control study in the WHI (BAA-24)—the Metabolomics of Coronary Heart Disease in the WHI with matching on 5-year age, race/ethnicity, hysterectomy status, and 2-year enrollment window [21]. Exclusion was described in Appendix A. ^b^ Metabolomics scores were derived from a linear combination of a series of elastic net regression models selecting biomarkers in relation to each energy-adjusted dietary score. ^c^ HRs were derived from multivariable-adjusted Cox proportional hazards regression models adjusted for the following baseline covariates: DM arm, if the participants were from a training dataset or a testing dataset, total energy intake, age at enrollment, physical activity, race and ethnicity, educational level, family history of cancer, number of hormones used, comorbidity score, baseline cardiovascular disease status, baseline lung disease, number of supplements used, non-steroidal anti-inflammatory drug use, hormone therapy study arm, baseline hormone therapy ever, oral contraceptive duration, pack years of smoking, coffee/tea and total alcohol intake. Colorectal cancer and subtype analyses were additionally adjusted for colorectal cancer screening. Invasive breast cancer and subtype analyses were additionally adjusted for baseline bilateral breast removal/benign breast diseases, hysterectomy age, months of breast-feeding, age at menopause, mammogram ever, parity, bilateral oophorectomy, passive smoking and Gail 5-year risk score. Endometrial cancer analyses were additionally adjusted for age at first birth, age at menarche, age at menopause, months of breast-feeding, and parity. Lung cancer analyses were additionally adjusted for baseline lung diseases, smoking status and passive smoking. ^d^ The *p* value for linear trend was estimated in the same multivariable-adjusted models by assigning the quintile-specific median value of each dietary pattern metabolomics score to all participants in the quintile and modelling as an ordinal variable. ^e^ P for continuous dietary score were calculated when entering the dietary metabolomics scores as a continuous variable into the models. ^f^ Diet-related metabolomics score beta coefficients were obtained from elastic net regression from the testing set.

## Data Availability

The data used in this project were provided by the Women’s Health Initiative. Data will be made available on request via the Women’s Health Initiative manuscript proposal process available at: https://www.whi.org/md/working-with-whi-data, accessed on 6 January 2020.

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
