# Peer review of "Hyperinsulinemic and Pro-Inflammatory Dietary Patterns and Metabolomic Profiles Are Associated with Increased Risk of Total and Site-Specific Cancers among Postmenopausal Women"

_cancers, 2023, doi:10.3390/cancers15061756_

Round 1
Reviewer 1 Report
This is an interesting analysis. The question is clinically relevant and conclusions are supported by the results. The manuscript is well written. A major limitation is the presentation of the results due to the length and the too much information on Tables/Figures. I support the publication of the manuscript, but I was wondering if the authors could trim some parts and emphasize the discussion to have a manuscript with a better flow. Otherwise, it looks good to me.
Author Response
This is an interesting analysis. The question is clinically relevant and conclusions are supported by the results. The manuscript is well written.
RESPONSE: Thank you for taking the time to review our manuscript and make suggestions for improvement
A major limitation is the presentation of the results due to the length and the too much information on Tables/Figures. I support the publication of the manuscript, but I was wondering if the authors could trim some parts and emphasize the discussion to have a manuscript with a better flow. Otherwise, it looks good to me.
RESPONSE: We agree with the reviewer and had difficult choices to make and how to present the large amount of results in a clear and easy to understand manner. We decided to make a graphical abstract with forest plots and selected a different color for the FDR-adjusted p-values deemed significant so that these can be easily identified while reading the figure. We also added the foods for each dietary pattern in color at the top, to help with understanding.
For the tables, we simplified the main results in table 2 to include mainly the number of cancer cases, incidence rate and relative risk for each cancer site, subsite or pathological type. The table could have been longer had we include age-adjusted values.
Reviewer 2 Report
In this work, the authors focus on an important issue relating dietary indexes with incidence of various cancer. This topic is of continued relevance as nutrition has been repeatedly pivotal in cancer prevention studies. Quantifying diet as the collaborators describe therefore serves an essential purpose in distinctly measuring the association between this factor and different cancers.
Thus, the reviewer recommends this work for publication given the following suggestions are addressed.
1. Have you thought of a p-value adjustment or lowering the level of significance.
2. May you include frequencies as well as percentages?
3. You may wish to reformat tables and make labels clearer.
Author Response
In this work, the authors focus on an important issue relating dietary indexes with incidence of various cancer. This topic is of continued relevance as nutrition has been repeatedly pivotal in cancer prevention studies. Quantifying diet as the collaborators describe therefore serves an essential purpose in distinctly measuring the association between this factor and different cancers. Thus, the reviewer recommends this work for publication given the following suggestions are addressed.
RESPONSE: We appreciate the reviewer time and effort to read and make recommendations for improving our manuscript.
1. Have you thought of a p-value adjustment or lowering the level of significance.
RESPONSE: Except for the metabolomics analyses, where we had limited sample size, we have adjusted our results for potential false discovery using the Benjamin-Hotchberg false discovery rate (FDR) adjustment. The FDR p values are listed in the figures and tables of the main results used to make conclusions.
2. May you include frequencies as well as percentages?
RESPONSE: In agreement with reviewer #1, we would prefer to keep the tables simpler as they are already quite long with a lot of data.
3. You may wish to reformat tables and make labels clearer.
RESPONSE: Please let us know the specific edits to include in the table/figure labels and we will be more than happy to include these.
Thank you.
Round 2
Reviewer 2 Report
Manuscript appears satisfactory for publication.